# Activity-dependent neuromodulation and calcium homeostasis cooperate to produce robust and modulable neuronal function

**Arthur Fyon**[ORCID]*, **Guillaume Drion**

Department of Electrical Engineering and Computer Science, University of Liège, Liège, Belgium

* afyon@uliege.be

## Abstract

Neurons rely on two interdependent mechanisms — homeostasis and neuromodulation — to maintain robust and adaptable functionality. Calcium homeostasis stabilizes neuronal activity by adjusting ionic conductances, whereas neuromodulation dynamically modifies ionic properties in response to external signals carried by neuromodulators. Combining these mechanisms in conductance-based models often produces unreliable outcomes, particularly when sharp neuromodulation interferes with calcium-homeostatic tuning. This study explores how a biologically inspired neuromodulation controller can harmonize with calcium homeostasis to ensure reliable neuronal function. Using computational models of stomatogastric ganglion and dopaminergic neurons, we demonstrate that controlled neuromodulation preserves neuronal firing patterns while calcium homeostasis simultaneously maintains target intracellular calcium levels. Unlike sharp neuromodulation, the neuromodulation controller integrates activity-dependent feedback through mechanisms mimicking G-protein-coupled receptor cascades. The interaction between these controllers critically depends on the existence of an intersection in conductance space, representing a balance between target calcium levels and neuromodulated firing patterns. Maximizing neuronal degeneracy enhances the likelihood of such intersections, enabling robust modulation and compensation for channel blockades. We further show that this controller pairing extends to network-level activity, reliably modulating the rhythmic activity of central pattern generators. This study highlights the complementary roles of calcium homeostasis and neuromodulation, proposing a unified control framework for maintaining robust and adaptive neural activity under physiological and pathological conditions.

## Author summary

Neurons must maintain stable activity while adapting to changing demands, relying on two mechanisms: calcium homeostasis, which stabilizes activity, and

**Data availability statement:** There are no primary data in the paper; all materials are available at https://github.com/arthur-fyon/BIOCONTROL_2025 and we have archived our code on Zenodo (DOI: 10.5281/zenodo.17827990).

**Funding:** AF is a Postdoctoral Researcher of the Fonds de la Recherche Scientifique - FNRS, supported by grant ASP-REN40024838 (to AF). AF received a salary from the FNRS under this grant. This work was additionally supported by the Belgian Government through the Federal Public Service Policy and Support, under grant NEMODEI2 (to AF and GD). The funders had no role in study design, data collection and analysis, decision to publish, or preparation of the manuscript.

**Competing interests:** The authors have declared that no competing interests exist.

neuromodulation, which adjusts behavior to external signals. These mechanisms often interact, but their improper coordination can lead to dysfunction. This study uses computational models to show how controlled neuromodulation — mimicking biological feedback — can be harmonized with calcium homeostasis to ensure robust neuronal function. The system compensates for channel blockades and maintains critical activity patterns for both single neurons and networks despite neuronal parameter variability. These insights not only deepen our understanding of neural robustness, but also suggest safer pharmacological strategies targeting neuromodulatory pathways, offering potential breakthroughs in treating neurological disorders without disrupting essential cellular mechanisms.

## Introduction

Brain activity is continuously shaped by neuromodulators and neuropeptides, including dopamine, serotonin, and histamine [1,2]. Neuromodulators dynamically influence single-neuron activity, input-output properties, and synaptic strength, enabling neuronal networks to adapt to changing needs, contexts, and environments [3–5]. This modulation occurs through the regulation of transmembrane proteins, such as ion channels and receptors, which alters neuronal excitability and synaptic dynamics [6,7]. While experimental studies highlight the ubiquity of neuromodulation, the fundamental principles governing its effects remain incompletely understood, making computational approaches crucial for unraveling its mechanisms.

Neuromodulation coexists with homeostatic plasticity, a process that gradually adjusts neuronal membrane properties to maintain a target activity level [8–14]. Neuronal cells are dependent on homeostatic mechanisms to maintain optimal functionality throughout the extended useful life of mammals [15]. Despite continuous turnover of transmembrane proteins, such as ion channels and receptors, occurring on varying temporal scales ranging from hours to weeks [16], the robustness and excitability of neurons must remain undisturbed [17]. Among homeostatic mechanisms, intrinsic homeostatic plasticity adjusts ion channel conductances in response to intracellular calcium levels [18,19]. However, since both neuromodulation and calcium homeostasis act on the same targets (ion channels) and may have competing objectives, understanding their interaction is essential [2,17].

We propose to study the interaction between neuromodulation and calcium homeostasis using conductance-based models. These models provide a biophysically accurate representation of neuronal dynamics by describing the electrical properties of the neuronal membrane. The membrane is modeled as an RC circuit, while ion channels are represented as nonlinear conductances that evolve according to experimentally derived activation and inactivation kinetics [20]. Conductance-based models remain the most faithful mathematical representations of biological neurons, as they allow direct incorporation of ion channel dynamics extracted from experimental data, following the methodology first introduced by Hodgkin and Huxley [21]. Over the years, numerous conductance-based models have been developed to describe a

variety of neuronal cell types across different species and brain regions [22–28]. This study focuses on a conductance-based model of stomatogastric ganglion (STG) neurons [22]. STG neurons are key components of rhythmic circuits that generate motor patterns driving stomach muscle contractions in crustaceans. This model was chosen for its ability to reproduce diverse firing patterns — ranging from tonic spiking to bursting — and its rich set of ion channels, which enable degeneracy. Specifically, the model includes classical fast sodium and delayed rectifier potassium channels, A-type and calcium-activated potassium channels, T-type and slow calcium channels, as well as hyperpolarization-activated H-channels.

We model intrinsic homeostasis using the calcium-homeostatic controller proposed by [29], which adjusts all ionic conductances based on the deviation of the neuron average intracellular calcium level from a target set-point. We refer to this mechanism as *calcium homeostasis* throughout this work, to distinguish it from other forms of homeostatic plasticity. Specifically, if the calcium level is lower than a target reference value, the controller increases ionic conductances according to calcium-homeostatic tuning rules, and conversely, decreases them if the calcium level is above the target. Importantly, these rules ensure that conductances are tuned along a single axis, preserving their relative ratios and maintaining correlations between them. This self-tuning mechanism allows neurons to counteract perturbations while preserving stable activity.

Neuromodulation is modeled using two different approaches. On the one hand, we model neuromodulation as a hard-wired change in target conductance values leading to a desired neuromodulated state in degenerate populations, which we call "sharp neuromodulation". On the other hand, we implement the neuromodulation controller developed by [30], which is designed to enable robust modulation of neuronal activity. We call this approach "controlled neuromodulation". One of its key properties is that it accounts for neuronal degeneracy, meaning that neurons with different ion channel compositions but similar activity patterns respond in a consistent manner to neuromodulation [31]. This feature is particularly important because biological neuronal populations exhibit significant variability in their underlying conductance parameters while maintaining functional stability. By leveraging degeneracy, the neuromodulation controller can apply the same control signal across a population of neurons with diverse conductance profiles, ensuring reliable and predictable neuromodulation.

We first describe the calcium-homeostatic controller and the two neuromodulation approaches, sharp and controlled, highlighting their distinct dependence on neuronal activity. We then investigate their interaction at the single-neuron level and show that combining intrinsic homeostasis with either sharp or controlled neuromodulation leads to vastly different outcomes. Next, we provide a mechanistic explanation for these different outcomes. We then extend our analysis to rhythmic neuronal networks, where neuromodulation plays a critical role in adjusting oscillatory patterns. By carefully integrating homeostatic and neuromodulatory control mechanisms, we aim to develop a unified framework that combines the strengths of both approaches. This integrated controller enables robust modulation of neuronal activity while ensuring long-term stability, shedding light on how neuromodulation and calcium homeostasis coexist in biological systems.

## Results

### Calcium homeostasis and neuromodulation: Two complementary regulatory mechanisms

We begin by describing the two regulatory mechanisms that are central to this study: calcium homeostasis and neuromodulation. Both mechanisms act on ion channel conductances, but they differ fundamentally in their objectives, timescales, and dependence on neuronal activity.

The calcium-homeostatic model [29] relies on intracellular calcium as a feedback signal, as illustrated in Fig 1A. The neuron continuously averages its intracellular calcium level over time and adjusts conductances accordingly: if calcium falls below a target level, ion channel transcription and translation increase, raising all conductance values, and vice versa. This phenomenon happens on a relatively slow timescale, which can last up to days [18]. Importantly, the calcium-homeostatic tuning rules ensure that conductances are tuned along a single axis, preserving their relative ratios

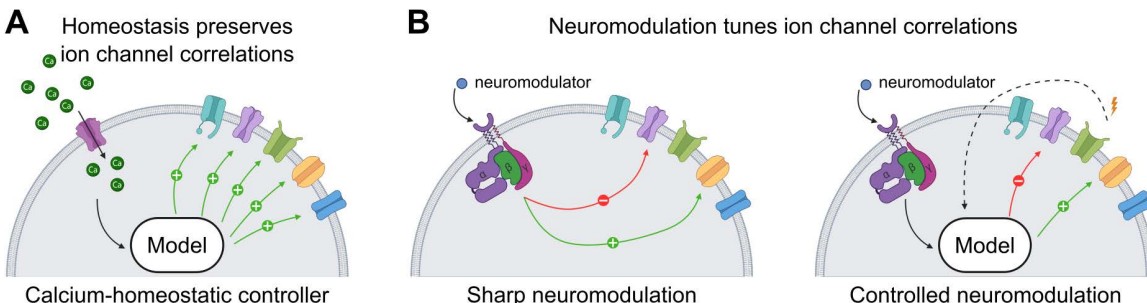

**Fig 1. Schematic representations of calcium homeostasis, sharp and controlled neuromodulation. A.** Schematic representation of calcium-homeostatic cascade. It senses intracellular calcium levels and regulates them by adjusting all conductances uniformly. **B.** Schematic representations of sharp (left) and controlled (right) neuromodulation. In sharp neuromodulation, neuromodulators bind to G-protein-coupled receptors, which directly affect subsets of ion channels independently of neuronal activity. In contrast, during controlled neuromodulation, G-protein-coupled receptors trigger complex signaling pathways that selectively affect subsets of ion channels in an activity-dependent way, hence producing a separate feedback loop over neuronal activity, acting in parallel with the calcium-homeostatic loop. Created in Bio Render. Fyon, A. (2026) https://BioRender.com/6q3irjy.

and maintaining correlations between them. Starting from any initial condition, this controller drives the cell along a line in conductance space known as the homogeneous scaling line. This self-tuning mechanism allows neurons to counteract perturbations while preserving stable activity.

In contrast, neuromodulation modifies a subset of ion channel conductances to reshape firing patterns. We consider two distinct implementations: sharp neuromodulation and controlled neuromodulation. *Sharp neuromodulation* is defined as a direct, hardwired, externally driven modification of a subset of ion channel maximal conductances, applied independently of neuronal activity (Fig 1B left). In this approach, neuromodulators bind to G-protein-coupled receptors, which directly affect subsets of ion channels without incorporating feedback from the neuron current activity state. Sharp neuromodulation, implemented using the method from [32], thus upsets the homeostasis setpoint, defined by the intracellular calcium target level. As calcium homeostasis subsequently acts to recover its calcium set point, there are no guarantees that the neuromodulated function is maintained as ion channel conductances are modified.

Alternatively, we can take into account that, in biological systems, neuromodulation occurs through an intracellular cascade that indirectly links changes in neuromodulator concentration and changes in targeted conductance values using intermediate signals such as, *e.g.*, cyclic adenosine monophosphate (cAMP) or intracellular calcium. We have recently shown that this indirect action of neuromodulation on conductance values could be viewed as an intracellular controller (Fig 1B right), which greatly improves the reliability of neuromodulatory action on highly degenerate populations [30]. We call this mechanism "controlled neuromodulation" to contrast with the sharp neuromodulation described above. Unlike sharp neuromodulation, which imposes activity-independent conductance changes, the neuromodulation controller provides continuous and adaptive modulation in an activity-dependent manner, ensuring stable functional transitions. This approach is biologically inspired, as there is experimental evidence that neuromodulation acts in an activity-dependent manner [2,33–37]. Specifically, the neuromodulation controller models the role of G-protein-coupled receptors and second messengers in a non-specific way, embedding feedback within the metabolic neuromodulation cascade [38,39]. During controlled neuromodulation, G-protein-coupled receptors trigger complex signaling pathways that selectively affect subsets of ion channels in an activity-dependent way, hence producing a separate feedback loop over neuronal activity, acting in parallel with the calcium-homeostatic loop.

At its core, the controller exploits Dynamic Input Conductances (DICs), a low-dimensional representation that links ion channel conductances to firing patterns (see Materials and methods for details). DICs reduce the high-dimensional space of ion channel conductances to three voltage-dependent feedback gains, each associated with a distinct timescale (fast,

slow, and ultraslow), whose values at threshold voltage reliably determine the firing pattern [40]. The controller operates in two stages. First, a feedforward step uses this linear mapping to compute neuron-specific reference values for the modulated conductances, given the desired firing pattern and the current state of all other (unmodulated) conductances. Second, a feedback step in the form of a Proportional-Integral (PI) controller tracks these references by regulating ion channel expression. Because the feedforward step depends on the full conductance state of the neuron, the controller naturally adapts its output to each individual neuron within a degenerate population, producing different conductance adjustments that all converge to the same target activity.

Importantly, the controller is agnostic to the specific neuromodulator identity and intracellular signaling cascade involved. This abstraction is justified by experimental evidence showing that diverse signaling pathways converge onto the same functional targets [2]. For instance, in the STG, multiple neuromodulators acting through distinct G-protein-coupled receptors and second messenger pathways (involving cAMP or calcium-dependent cascades) all converge onto the same modulator-activated inward current $I_{MI}$ [41]. In this context, the feedforward step captures the dependence of neuromodulatory effects on the current activity state of the neuron [33], while the feedback step (PI controller) models the temporal integration inherent to intracellular signaling cascades, where second messenger concentrations gradually adjust in response to sustained receptor activation. Rather than modeling any particular cascade, our controller captures the functional principle common to these diverse pathways: an activity-dependent feedback loop that senses neuronal output and adjusts a targeted subset of conductances to reach a desired activity pattern.

## Sharp neuromodulation interferes with calcium homeostasis whereas controlled neuromodulation naturally leads to robust and modulable neuronal function

Having established the two neuromodulation approaches, we now examine their interaction with calcium homeostasis. We specifically focus on a robust transition from tonic spiking to bursting in degenerate neuronal populations, *i.e.,* populations composed of neurons that exhibit similar firing behaviors despite differences in underlying conductance profiles. We applied both types of neuromodulation to a population of degenerate neurons generated using the dataset from [32], modifying the maximal conductances of A-type potassium ($\bar{g}_A$) and slow calcium ($\bar{g}_{CaS}$) channels to transition neurons from tonic spiking to bursting, and then allowed calcium-homeostatic compensation to unfold. In all illustrative simulations (Fig 2), the time constant of the calcium-homeostatic controller was reduced so that the dynamics of both neuromodulation and calcium-homeostatic compensation could be observed within a single trace, without affecting the qualitative behavior of the system.

We first test the interaction of calcium homeostasis and sharp neuromodulation (Fig 2A-C). All conductance trajectories ($N=200$) are shown in Fig 2A, while Fig 2B illustrates calcium dynamics across the population. Specific time evolution curves for single neurons within the population can be found in Supporting Information, S1 Appendix. Despite initially similar firing patterns ($t_{before}$) and a similar outcome of sharp neuromodulation ($t_{after}$), calcium-homeostatic compensation produced highly variable and often unreliable outcomes in terms of modulated activity ($t_{compensated}$), while still regulating intracellular calcium (Fig 2C). Depending on the initial conductance state, neuromodulation led to new states where calcium homeostasis could either maintain functional activity, result in pathological behavior (loss of bursting, *i.e.,* a burstiness of 0), or drive neurons into unphysiological regimes. This finding underscores a fundamental property of degeneracy: neurons with different underlying conductances can exhibit similar activity yet respond differently to perturbations, such as sharp neuromodulation. Note that these differences in channel conductances are consistent with biological measurements: different channel types have effective conductances spanning several orders of magnitude [29,43]. Moreover, the effective conductance of a given ion channel type can vary severalfold within a single neuron type [44]. The variability in the dataset from [32] is inspired by these biological measurements and slightly exceeds the observed range to test the robustness of our analysis.

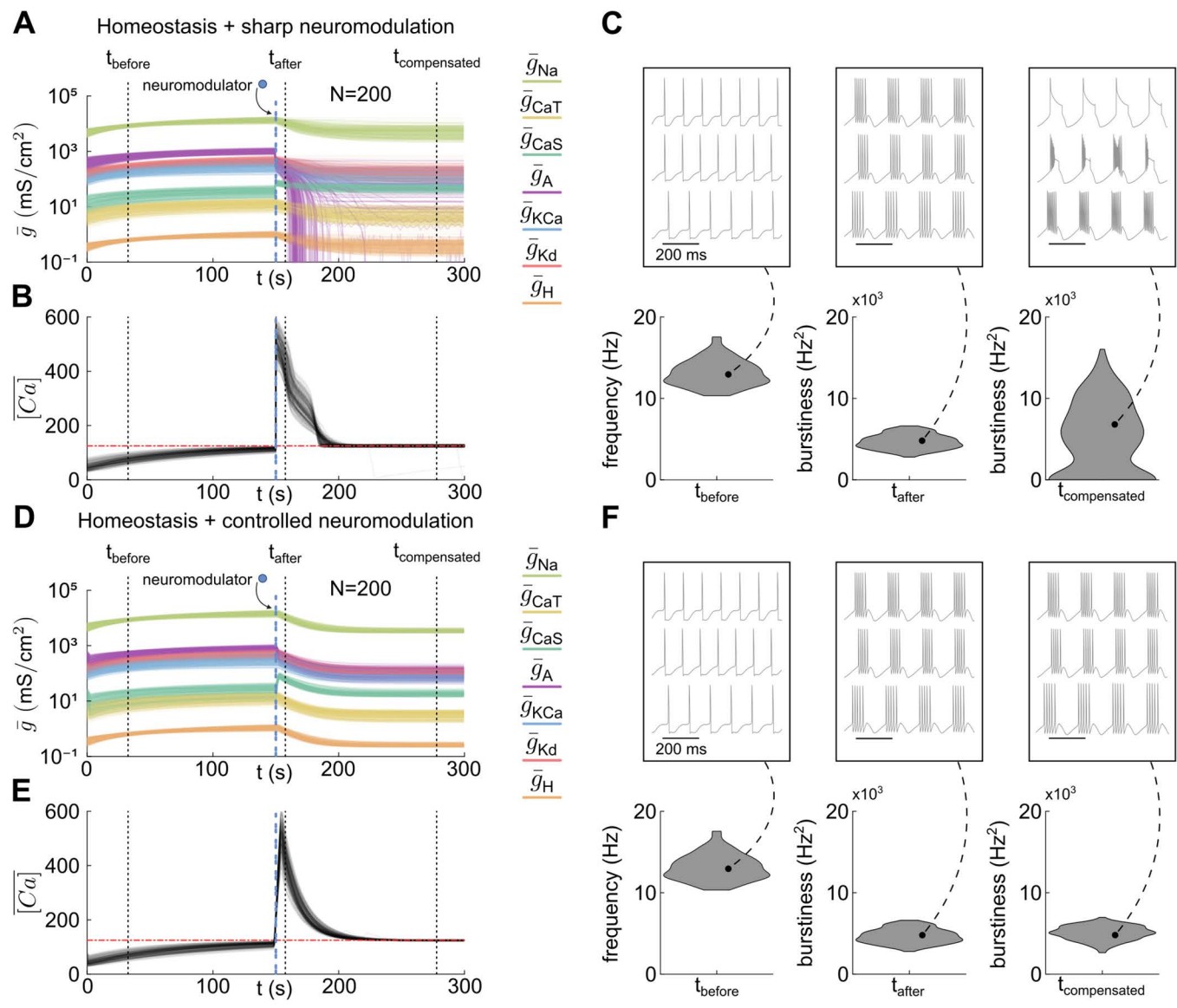

**Fig 2. Sharp neuromodulation interferes with homeostasis whereas controlled neuromodulation leads to robust and modulable neuronal function. A.** Time evolution of all conductances in the STG model, displayed on a logarithmic scale, during calcium homeostasis with sharp neuromodulation (instantaneous changes in $\bar{g}_{CaS}$ and $\bar{g}_A$ starting at the dashed blue line) for a degenerate population of $N = 200$ neuron models. **B.** The corresponding mean intracellular calcium concentration over time shows the target value (red dash-dotted line) is maintained both before and after sharp neuromodulation. **C.** Three representative traces of neurons within the population are highlighted (top) before neuromodulation ($t_{before}$), right after ($t_{after}$) and after calcium-homeostatic compensation ($t_{compensated}$); along with the activity distributions of the population at these three time points (bottom). On the one hand, before neuromodulation, the population exhibits tonic firing, and hence the firing frequency (defined as the reciprocal of the interspike interval) is used as the activity metric. On the other hand, after neuromodulation, the population exhibits bursting, and hence the burstiness is used as the activity metric. Burstiness is defined as in [42], *i.e.*, the product of the intraburst frequency (reciprocal of the interspike interval within a burst), the interburst frequency (reciprocal of the interburst interval), and the number of spikes per burst. A burstiness of zero means that the neuron is not producing multi-spike bursts (silence, tonic firing, or single-spike bursters). **D.** Same as panel A, but with controlled neuromodulation (controlled changes in $\bar{g}_{CaS}$ and $\bar{g}_A$ starting at the dashed blue line). **E.** Same as panel B, but with controlled neuromodulation. **F.** Same as panel C, but with controlled neuromodulation.

This type of interference was anticipated in [29], where the authors demonstrated that calcium-homeostatic regulation can produce diverse responses to perturbations such as ion channel deletions. Depending on the initial conditions, a deletion may either preserve function, disrupt it, or lead to pathological compensation. Our results extend this idea by showing that even physiological disturbances, such as neuromodulation, can induce similar unpredictability when coupled with calcium-homeostatic regulation.

Using controlled neuromodulation on the same population dramatically changes the outcome (Fig 2D-F). Fig 2D-E illustrates the transition from tonic spiking to bursting in the same degenerate population of STG neuron models as in Fig 2A-C. Specific time evolution curves for single neurons within the population can be found in Supporting Information, S1 Appendix. Initially, all neurons exhibit tonic firing, with calcium homeostasis tuning conductances along a tonic spiking homogeneous scaling line to maintain a target calcium level ($t_{before}$ in Fig 2F). Midway through the simulation, controlled neuromodulation is applied, controlling $\bar{g}_{CaS}$ and $\bar{g}_A$ to induce bursting ($t_{after}$ in Fig 2F). As in the case of sharp neuromodulation, this transition increases intracellular calcium levels, leading to an adaptive shift in conductance values. Calcium-homeostatic regulation then readjusts overall conductance magnitudes, but here the interaction between calcium homeostasis and controlled neuromodulation ensures that the newly established neuromodulated state is maintained. Ultimately, the population reaches a stable configuration where both the neuromodulated function and intracellular calcium homeostasis are fulfilled ($t_{compensated}$ in Fig 2F).

The differences between sharp and controlled neuromodulation can also be highlighted through neuromodulator washout, *i.e.*, removal of the neuromodulatory input. At both $t_{after}$ and $t_{compensated}$, washout recovers tonic spiking across the population under both sharp and controlled neuromodulation, though for fundamentally different reasons and with different levels of fidelity. Under sharp neuromodulation, calcium-homeostatic compensation continuously drives conductances along the native tonic spiking scaling direction, so that tonic spiking is the only remaining stable activity pattern. Recovery of spiking upon washout is therefore expected, as the system has no other stable attractor. However, when washout occurs at $t_{compensated}$, the conductance distribution of the population does not exactly match its pre-neuromodulation state: because calcium homeostasis has been compensating for the sharp neuromodulatory perturbation throughout, the resulting conductance profiles are subtly altered, leading to a slightly different distribution of firing frequencies across the population. Although the qualitative behavior (tonic spiking) is preserved, this quantitative drift implies that repeated cycles of sharp neuromodulation and washout could progressively erode the fidelity of neuronal function, as each cycle introduces small but cumulative changes in the conductance landscape. In contrast, under controlled neuromodulation, the system can maintain any target activity through activity-dependent feedback. Upon washout, the neuromodulatory reference is removed, and the system returns to tonic spiking because the calcium-homeostatic controller regains sole authority over conductance tuning. Because the controlled neuromodulation mechanism continuously coordinates with calcium homeostasis rather than competing with it, the pre-neuromodulation conductance distribution is faithfully recovered upon washout.

Controlled neuromodulation tunes the direction of calcium homeostasis-driven changes in conductance values to ensure stable neuromodulated function during calcium-homeostatic regulation

To study the mechanisms underlying the constructive interaction between controlled neuromodulation and calcium homeostasis, we plot the conductance trajectories of the simulations from Fig 2 in the plane of modulated conductances, $\bar{g}_{CaS}$ and $\bar{g}_A$, during tonic spiking calcium-homeostatic compensation for both sharp and controlled neuromodulation (Fig 3A). Initially, as calcium levels remain below the threshold, calcium homeostasis increases the modulated conductances along a homogeneous scaling direction characteristic of tonic spiking. This direction corresponds to a line intersecting the origin, with variability due to conductance degeneracy within the population, and its slope is determined by the relative time constants of mRNA transcription in the calcium homeostasis model, with steady-state analysis predicting $\bar{g}_i/\bar{g}_j \approx \tau_{m_j}/\tau_{m_i}$ [29]. When neuromodulation is applied, all conductances are adjusted to reshape their ratios and induce bursting (Fig 3B). Importantly, neuromodulatory actions follow the same direction across neurons but vary in magnitude depending on

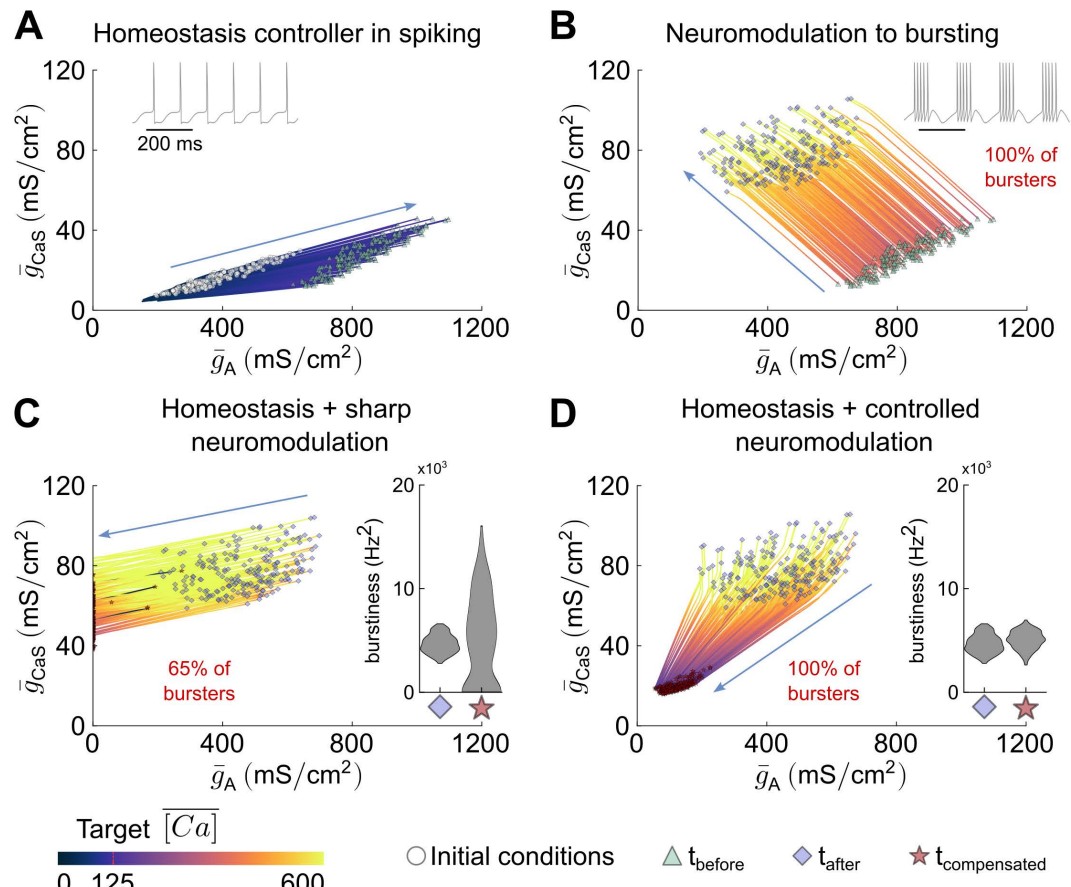

**Fig 3. Controlled neuromodulation tunes the direction of homeostasis-driven changes to ensure stable neuromodulated function. A.** Trajectories of the STG population from Fig 2 in the modulated conductance space during homogeneous scaling in tonic spiking only. As expected from calcium homeostasis, all neurons move in the same direction. White circles represent the initial conditions at the start of the step, and the green triangle marks the final conditions. **B.** Same as panel A, but during the brief period when neuromodulation is active. All trajectories are parallel to one another. **C.** Same as panels A and B, but during the calcium homeostasis compensation phase with sharp neuromodulation, resulting in non-robust outcomes. **D.** Same as panel C, but with controlled neuromodulation, resulting in function-preserving robust outcomes.

individual neuronal states [32]. This modulation leads to a sharp increase in intracellular calcium levels. After neuromodulation is applied, the long-term outcome depends on the type of neuromodulation used (Fig 3C-D).

Under sharp neuromodulation, calcium homeostasis operates alone in the bursting phase, following the scaling direction characteristic of its reference, *i.e.,* unmodulated, state set by the relative time constants of mRNA transcription in the calcium homeostasis model (Fig 3C). This results in a decrease in calcium levels, but as shown in Fig 2A-C, the response is unreliable and often undesirable, resulting in a loss of robust bursting across the population (bursting is either lost entirely, sustained with substantially altered properties, or replaced by irregular activity). Specifically, for most neurons, the calcium-homeostatic controller reduces $\bar{g}_A$ toward negative values (saturated at 0 to remain in physiological ranges and prevent model instability) to compensate for $\bar{g}_{CaS}$, driving the neuron away from robust bursting and leading to heterogeneous activity changes across the population (violin plots in Fig 3C).

In contrast, controlled neuromodulation allows the homeostatic and neuromodulation controllers to act together, maintaining the new conductance ratios associated with bursting (Fig 3D). As a result, calcium-homeostatic regulation follows a new homogeneous scaling trajectory aligned with bursting conductance ratios. These trajectories converge toward the

origin, as in tonic spiking, but with a distinct direction determined by neuromodulation. Due to initial conductance degeneracy, the exact scaling varies across the population. The shift between the two homogeneous scalings — tonic spiking and bursting — emerges from neuromodulation, ensuring that the desired neuromodulated function is maintained (violin plots in Fig 3D).

The incompatibility of sharp neuromodulation with calcium homeostasis can be further analyzed using a schematic representation of the modulated conductance plane (Fig 4A). In this plane, different combinations of $\bar{g}_{CaS}$ and $\bar{g}_A$ can yield similar calcium levels, forming calcium isoclines (green dashed lines). Similarly, different conductance values can result in similar firing activity, forming activity isoclines (blue dashed lines). While calcium isoclines are parallel, activity isoclines radiate from the origin and rotate around it. This organization reflects calcium-homeostatic tuning rules: different conductance ratios correspond to distinct firing modes [29]. Initially, the neuron operates at the intersection of a target spiking isocline and a target calcium isocline (at $t_{before}$). When sharp neuromodulation is applied (black arrow in Fig 4A), conductances shift to a new state corresponding to strong bursting, moving the neuron to a higher calcium isocline. Calcium homeostasis then acts to restore calcium levels (solid red arrow in Fig 4A), but because it retains the "native" tonic spiking scaling direction (dashed red arrow in Fig 4A), it alters conductances along the spiking isocline rather than preserving the newly induced bursting state. As a result, the neuron drifts away from the bursting isocline, potentially disrupting its function. This explains why sharp neuromodulation leads to unreliable outcomes when coupled with calcium homeostasis.

To reconcile calcium homeostasis and neuromodulation and achieve robust functional modulation, we leverage their distinct timescales. Neuromodulation controls specific ionic conductances, while calcium homeostasis adjusts all conductances over a slower timescale to regulate calcium levels. In the modulated conductance plane (Fig 4B), the neuron initially follows the same trajectory as in Fig 4A. However, under controlled neuromodulation, both controllers remain active in an activity-dependent way. When calcium homeostasis lowers calcium levels and shifts the neuron away from the bursting isocline, the neuromodulatory controller counteracts this deviation, bringing the neuron back toward the bursting

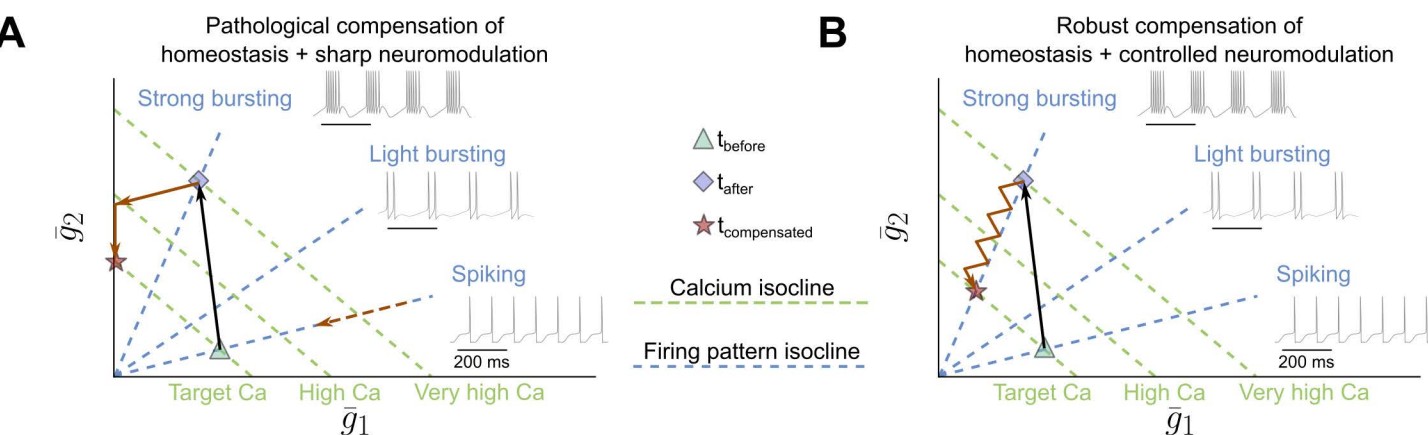

**Fig 4. Schematic explanation of why controlled neuromodulation integrates better with homeostasis compared to sharp neuromodulation.**
**A.** Schematic of the modulated conductance space for calcium homeostasis combined with sharp neuromodulation. Initially at $t_{before}$ (green triangles), the neuron spikes at the target calcium level, located at the intersection of the firing pattern isocline (dashed blue lines) and the calcium level isocline (dashed green lines). Following sharp neuromodulation (black arrow) to a new firing pattern at $t_{after}$ (blue diamond), calcium levels sharply increase. Homeostasis reduces calcium (solid red arrow) by moving along the same direction as before neuromodulation (dashed red arrow), causing the neuron to deviate from the neuromodulated firing pattern isocline at $t_{compensated}$ (red star). **B.** Same as panel A, but with controlled neuromodulation. The initial steps are identical, but after neuromodulation, as calcium homeostasis acts to reduce calcium, controlled neuromodulation simultaneously adjusts to keep the neuron on the neuromodulated firing pattern isocline (sawtooth red arrow), as controlled neuromodulation is activity-dependent. Because controlled neuromodulation operates on a much faster timescale than calcium homeostasis, the neuron remains on the neuromodulated firing pattern isocline throughout. Black arrow: trajectory of sharp (A) or controlled (B) neuromodulation. Solid red arrow: trajectory of calcium-homeostatic compensation. Dashed red arrow: direction of the calcium-homeostatic compensation in tonic firing (A).

isocline and preserving the desired function. Because the neuromodulatory controller operates orders of magnitude faster than the calcium-homeostatic controller, its rapid corrections render homeostatic perturbations negligible. This interaction constrains changes in conductance ratios along the target activity isocline (strong bursting in this case). Ultimately, the neuron stabilizes at the intersection of the target calcium and target activity isoclines, where both controllers achieve their objectives.

The results above were obtained with controlled neuromodulation acting on A-type potassium ($\bar{g}_A$) and slow calcium ($\bar{g}_{CaS}$) conductances. However, because the neuromodulation controller is agnostic to the specific neuromodulator and signaling cascade, we also conducted experiments in which H-type channels ($\bar{g}_H$) are modulated, given that these channels are well known to be sensitive to cAMP levels [45]. An analogous figure to Fig 2, with neuromodulation acting on H-type channels, can be found in Supporting Information, S1 Appendix. These experiments yield similar results: failure under sharp neuromodulation and robust function under controlled neuromodulation. This confirms that the observations are not specific to the particular subset of modulated channels, but rather emerge from the activity-dependent control action itself.

## Controlled neuromodulation and calcium homeostasis ensure the preservation of function under physiologically recoverable disturbances

By employing this tandem of controllers, neuronal function becomes robust to channel blockade, provided that the channel deletion is compensable. While calcium homeostasis alone may lead to unreliable recovery, incorporating controlled neuromodulation stabilizes the neuronal response to blockade, even in highly degenerate neuronal populations. Furthermore, if the function of the blocked channel can be physiologically recovered, meaning that other channels can substitute for its role, there exists an optimal combination of neuronal feedback gains (*i.e.*, controlled neuromodulation input) that preserves the function. In essence, when a channel is blocked and its function is compensable, a specific neuromodulator concentration (or cocktail of neuromodulators) can maintain the neuronal function.

For example, in Fig 5, various channel blockades are applied to a degenerate population of STG conductance-based models equipped with the combined calcium homeostasis/neuromodulation tandem. The target behavior of the population is regular bursting. In Fig 5A, H-type channels are blocked. Here, the function remains intact even immediately after the blockade, without requiring compensation (violin plots in Fig 5A right). The combined action of calcium homeostasis and controlled neuromodulation then ensures that intracellular calcium levels are restored on a population-wide scale, maintaining the function.

In Fig 5B and C, T-type calcium and calcium-activated potassium channels are blocked, respectively. In these cases, the blockade largely disrupts bursting (violin plots in Fig 5B-C right). However, allowing the calcium homeostasis/neuromodulation controllers to compensate restores function in most neurons, albeit with some displaying irregular bursting or reduced burstiness following compensation for the KCa channel blockade (violin plot in Fig 5C far right). Additionally, intracellular calcium levels stabilize at their target values. Despite operating at different timescales and membrane potentials, and thus producing different acute physiological effects upon blockade, recovery is possible for these channels because their functions overlap with those of other channels in the model, *i.e.*, they exhibit degeneracy [31]. The STG neuron model includes numerous channels operating on slow and ultraslow timescales, enabling compensation over these timescales. The function of calcium-activated potassium channels is less degenerate: its role cannot be fully compensated, as reflected in the reduced population burstiness, although the bursting function itself is maintained. However, even when the blockade is theoretically compensable, the control input may need to be adjusted. In the case of the KCa channel blockade, the loss was compensated by tuning the neuronal feedback gains, which corresponds to applying a different neuromodulator concentration to partially recover the function. This highlights that, even when compensation is theoretically possible, the neuron may not reach the compensated state without adjusting the neuromodulator cocktail.

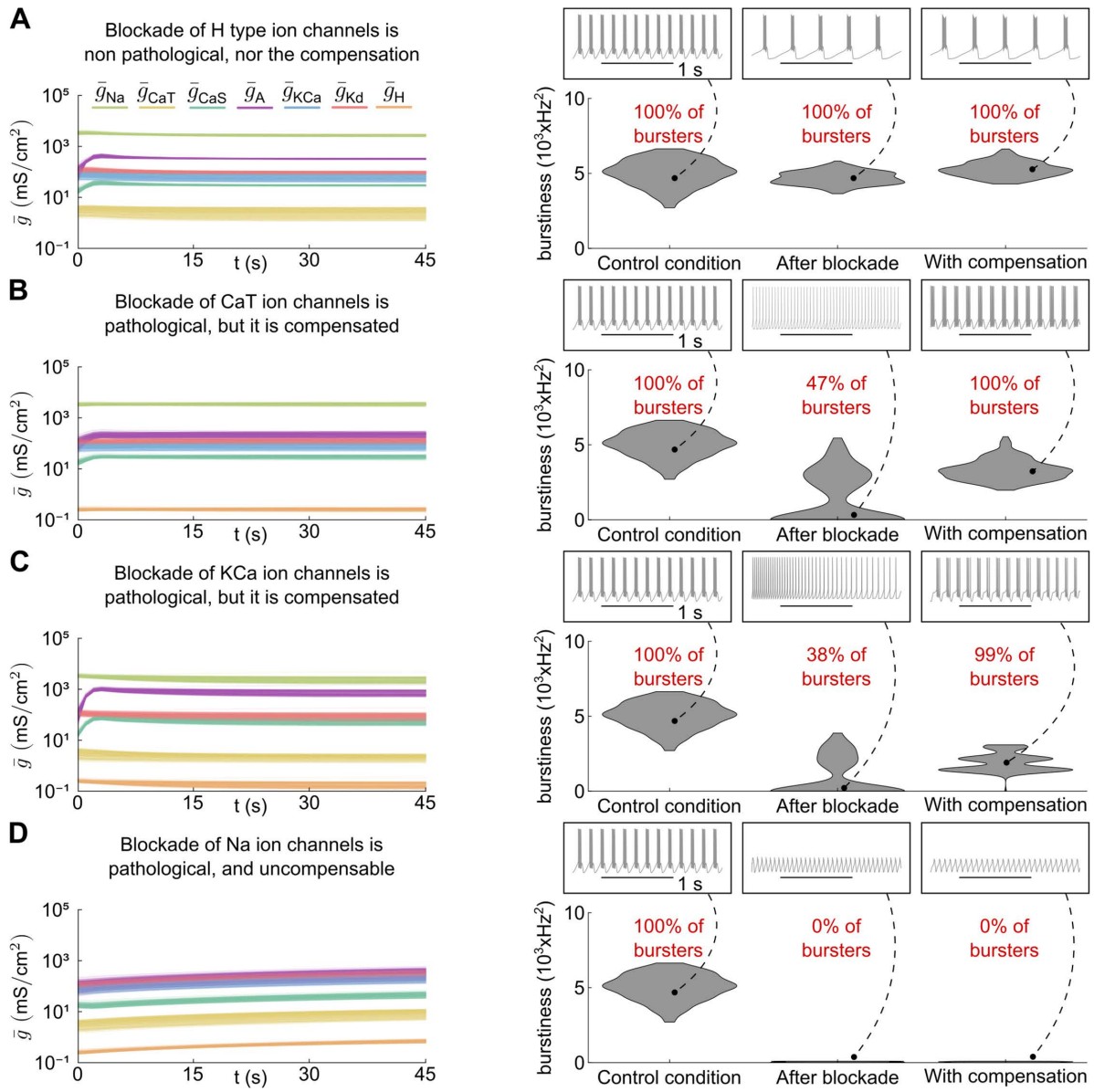

**Fig 5. Controlled neuromodulation and homeostasis ensure the preservation of function under physiologically recoverable disturbances. A.** Time evolution of all conductances in the STG model, displayed on a logarithmic scale, during calcium homeostasis with controlled neuromodulation (controlled changes in $\bar{g}_{CaS}$ and $\bar{g}_A$) for a degenerate population of $N=200$ neuron models with H-type ion channel blockade (left). The corresponding distribution of behaviors before the blockade, immediately after the blockade, and following compensation (right) demonstrates the effect of the blockade. In this case, no neurons lose bursting due to the channel blockade or compensation. **B.** Same as panel A, but with a CaT channel blockade. The blockade causes half of the neurons to lose bursting; however, all neurons recover bursting with compensation. **C.** Same as panel A, but with a KCa channel blockade. The blockade causes more than half of the neurons to lose bursting, but nearly all recover bursting with compensation, although the burstiness of the population remains altered. **D.** Same as panel A, but with a Na channel blockade. The blockade causes all neurons to lose bursting, and compensation does not restore bursting due to the non-degenerate essential role of Na channels.

In contrast, the blockade of fast transient sodium channels is irrecoverable (Fig 5D), as these channels are essential for spike initiation on a fast timescale. Once the blockade is applied, compensation cannot restore spiking because sodium channel function is non-degenerate and operates on a timescale that alternative channels cannot compensate for.

Consequently, intracellular calcium levels also fail to reach their target values. A similar outcome would occur following the blockade of delayed rectifier potassium channels, which are responsible for the downstroke of spikes.

## Central pattern generator networks can be robustly modulated to orchestrate their rhythmic output

The application of this tandem of controllers extends beyond single neurons to networks, particularly central pattern generators (CPGs). CPGs are self-organized neuronal circuits composed primarily of bursting neurons connected through inhibitory synapses, capable of generating rhythmic output without external input [46,47]. A well-studied example is the circuit responsible for the pyloric and gastric mill rhythms in crustaceans, located in the STG. This network produces two distinct rhythms: the fast pyloric rhythm and the neuromodulated slow gastric mill rhythm. These rhythmic outputs are essential for coordinating muscle contractions in the stomach for digestion, among other functions. A minimal network model of such a bi-rhythmic circuit is investigated in Fig 6 [48,49]. It consists of two half-center oscillators operating at different speeds, connected via a central neuron. Half-center oscillators are formed by two bursting neurons connected through mutual inhibition, allowing them to fire anti-phase bursts. In this model, the purple and blue neurons constitute the

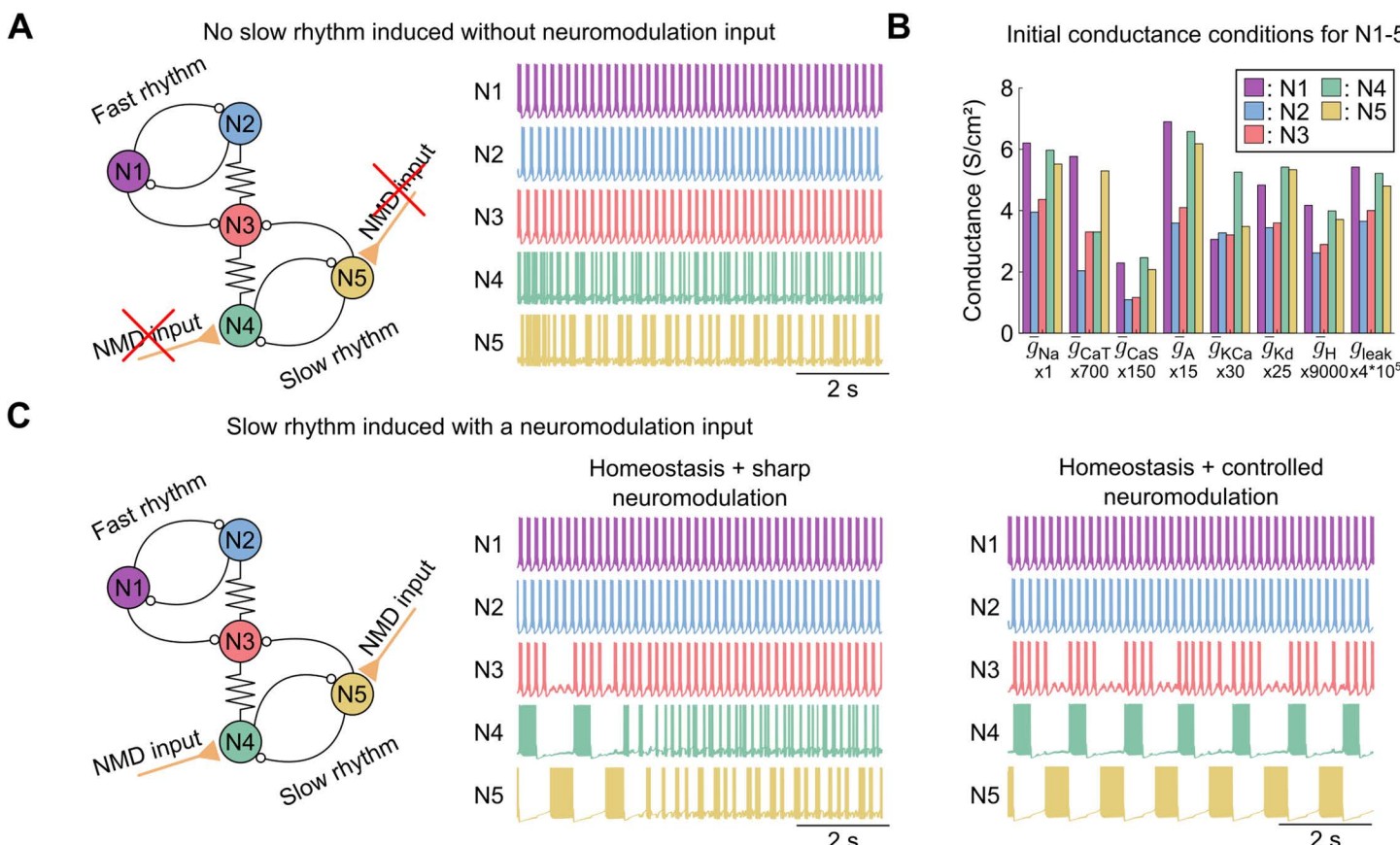

**Fig 6. Central pattern generator networks can be robustly modulated to orchestrate their rhythmic output. A.** Control condition of the simplified pyloric/gastric mill rhythm network composed of 5 neurons (N1-N5) (left), inspired by [48]. Without neuromodulation, the fast rhythm is active (top two traces, middle), while the slow rhythm is inactive (bottom two traces, middle). **B.** Initial conductance values for the five neurons of the network (N1-N5, color coded). Multiplier coefficients have been applied to all conductances to facilitate visualization in a non-logarithmic graph, allowing direct comparison of the different initial conductance values across all neurons, as in [43]. **C.** Schematic representation of the activated slow rhythm network (left), modulated by either sharp neuromodulation (middle) or controlled neuromodulation (right). With sharp neuromodulation, the slow rhythm fails to be sustained due to its unreliability, whereas controlled neuromodulation sustains the slow rhythm.

fast half-center oscillator (representing the pyloric rhythm), while the green and yellow neurons form the neuromodulated slow half-center oscillator (representing the gastric mill rhythm). The red neuron acts as the central element, also in bursting mode. It has been shown that the gastric mill rhythm requires neuromodulatory input to be activated [50].

In this study, the tandem of controllers reliably induces the gastric mill rhythm, whereas calcium homeostasis combined with sharp neuromodulation does not. Under control conditions (Fig 6A), *i.e.,* in the absence of neuromodulatory input to the gastric mill rhythm, only the pyloric rhythm remains active, with the central neuron bursting continuously. Notably, variability was introduced in the initial conductance values across all neurons of the network to assess the robustness of the controllers in the presence of degeneracy (Fig 6B). When neurons are equipped with the calcium homeostatic controller only, the activation of sharp neuromodulation for the gastric mill rhythm leads to a rapid loss of function due to the unreliability of this combination (Fig 6C middle). However, with controlled neuromodulation, the gastric mill rhythm is successfully induced and sustained (Fig 6C right). Despite the intrinsic degeneracy of the neurons, reflected in significant variability in conductance ratios, the network achieves stable rhythmic output while maintaining appropriate intracellular calcium levels. The mechanisms underlying these contrasting outcomes at the network level directly parallel those identified in the single-neuron analysis, as detailed below.

More specifically, under sharp neuromodulation combined with calcium homeostasis (Fig 6C middle), the slow rhythm initially emerges but rapidly deteriorates as calcium-homeostatic compensation unfolds. As in the single-neuron case (Fig 4A), calcium homeostasis drives conductances along the native, unmodulated scaling direction, progressively pulling the slow-rhythm neurons (N4 and N5, green and yellow) away from the bursting isocline required to sustain anti-phase oscillations. The half-center oscillator mechanism critically depends on both neurons maintaining robust bursting with appropriate duty cycles. Consequently, the loss of bursting in even a single neuron is sufficient to collapse the entire slow rhythm. In contrast, under controlled neuromodulation (Fig 6C right), the activity-dependent feedback continuously corrects for homeostatic drift, maintaining each neuron on its target bursting isocline as described in Fig 4B. This ensures that the slow-rhythm neurons preserve the conductance ratios necessary for sustained anti-phase bursting, while the central neuron (N3, red) appropriately alternates between the fast and slow rhythmic patterns. Importantly, the initial conductance variability across all five neurons (Fig 6B) confirms that these network-level results are robust to degeneracy: despite substantial differences in individual conductance profiles, the tandem of controllers successfully coordinates population-wide rhythmic output. The fast rhythm (N1 and N2, purple and blue) remains active throughout, with only minor phase adjustments resulting from the electrical coupling to the slow-rhythm neurons. This is consistent with the experimental observation that pyloric rhythm modulation during gastric mill activation mainly involves changes in phase relationships [2], confirming that neuromodulation here primarily acts to activate or deactivate the slow rhythm.

This toy experiment demonstrates that the tandem of calcium homeostasis and controlled neuromodulation can be applied in computational models to modulate and sustain not only single-neuron activity but also network-level dynamics.

## Discussion

The efficient and robust integration of calcium homeostasis and neuromodulation in neurons remains an open question, with several hypotheses proposed to explain their interaction [2,51]. Improper integration of these mechanisms, for instance combining sharp neuromodulation with a calcium-homeostatic controller, can lead to negative interactions, resulting in unreliable behaviors or numerical instabilities.

In contrast, pairing a calcium-homeostatic controller with controlled neuromodulation enables reliable modulation of neural function while maintaining appropriate calcium levels. This approach is biologically inspired, as neuromodulation is known to be activity-dependent, as observed experimentally [33–35]. This reliable interaction extends to neural circuits, such as the gastric mill circuit in crabs, where it contributes to robust network-level function [2].

However, the success of this interaction critically depends on a point in the conductance space where both controllers converge. This point represents a neural activity pattern that supports neuromodulated firing while maintaining

homeostatic calcium levels. As shown in Fig 4, it corresponds to the intersection of the strong bursting isocline (the target of controlled neuromodulation) and the homeostatic calcium setpoint. When no such intersection exists — *e.g.*, under sodium channel blockade (Fig 5D) — neither neuromodulated activity nor calcium homeostasis can be sustained. This may lead to pathological compensations, where one controller prioritizes calcium regulation at the expense of the target firing pattern [52–54]. Additionally, discontinuities in neuromodulated firing patterns may induce transient pathological behaviors. Moreover, the mere existence of such an intersection point does not guarantee that a neuron will reach it. For instance, to compensate for the loss of KCa channels (Fig 5C), a different neuromodulation target was required, corresponding to a different neuromodulator concentration. This means that, even when the intersection point exists, the neuron may require a specific neuromodulator concentration (or cocktail of neuromodulators) to partially or fully restore its function.

Maximizing convergence between these control mechanisms requires increasing the probability of both an intersection and a continuous path in conductance space. We hypothesize that this can be achieved by maximizing neuronal degeneracy [31,55–58]. Higher degeneracy increases the likelihood that the functional and dynamical role of one ion channel can be compensated by others, thereby preserving both the path and the intersection. This highlights the crucial role of degeneracy in enabling robust, neuromodulated neural function.

To further demonstrate the generality of the proposed framework, we applied it to a midbrain dopaminergic neuron model adapted from [23]. As in the STG model, a transition from tonic spiking (pacemaking) to strong bursting was induced by modulating N-type and L-type calcium conductances ($\bar{g}_{CaN}$ and $\bar{g}_{CaL}$) through controlled neuromodulation. Calcium-homeostatic compensation continuously regulated calcium levels, ensuring a reliable and robust outcome despite initial conductance degeneracy within the neuronal population (see Supporting Information, S1 Appendix, for details).

These constraints on compensation have direct implications for understanding neurological and neurodegenerative conditions. Our framework predicts that the clinical severity of channelopathies, in which mutations impair or eliminate specific ion channel functions, would depend on the degree of degeneracy of the affected channel [59]: mutations affecting highly degenerate channels may be largely compensated and produce mild phenotypes, whereas mutations affecting non-degenerate channels would lead to severe dysfunction. Similarly, the progressive loss of neuromodulatory input characteristic of neurodegenerative diseases, such as dopamine depletion in Parkinson's disease [60], can be interpreted within our framework as a gradual loss of the controlled neuromodulation component. In this scenario, the calcium-homeostatic controller would increasingly operate alone, potentially producing the kind of pathological compensations observed in our sharp neuromodulation results (Fig 2A-C). This perspective suggests that therapeutic strategies aimed at preserving or restoring the activity-dependent feedback loop, rather than directly replacing the lost neuromodulator, may be more effective at maintaining robust neuronal function.

These findings underscore the potential risks associated with pharmacological ion channel blockade [59,61,62]. If neuronal degeneracy is insufficient (or if the amount of neuromodulator released is unchanged), such interventions may compromise the ability to sustain robust neuromodulation. As an alternative, we propose targeting elements within the neuromodulation cascade — such as second messengers or other components of the control loop — as more reliable pharmacological strategies. By preserving the dynamics of controlled neuromodulation, these interventions could offer a more robust alternative to direct ion channel blockade.

Finally, the controlled neuromodulation framework presented here operates on single-compartment models. Extending it to multi-compartment models with dendritic conductances is a natural direction for future work. In principle, the DIC framework generalizes to spatially extended models, as the sensitivity matrix can be augmented to include dendritic conductances and electrotonic coupling terms [63]. However, this extension raises challenges related to the increased dimensionality of the conductance space and the established non-uniformity of dendritic ion channel distributions [64]. Furthermore, recent experimental evidence demonstrates that neuromodulators can act in a compartment-specific manner [65], and computational studies show that the same neuromodulatory action can have opposing effects at different

dendritic locations [66]. These observations suggest that spatially distributed or hierarchical control architectures may be required. Addressing these challenges in the context of controlled neuromodulation remains an open direction for future work.

## Materials and methods

### Programming language

The Julia programming language was used in this work [67]. Numerical integration was realized using *DifferentialEquations.jl*.

### Conductance-based model

For all experiments, single-compartment conductance-based models were employed. These models articulate an ordinary differential equation for the membrane voltage $V$, where $N$ ion channels are characterized as nonlinear dynamic conductances, and the phospholipid bilayer is represented as a passive resistor-capacitor circuit. Mathematically, the voltage-current relationship of any conductance-based neuron model is expressed as follows:

$$I_C = C\frac{dV}{dt} + g_{leak}(V - E_{leak}) = -I_{int} + I_{ext},$$
$$= -\sum_{ion \in \mathcal{I}} g_{ion}(V, t)(V - E_{ion}) + I_{ext}.$$

Here, $C$ represents the membrane capacitance, $g_{ion}$ denotes the considered ion channel conductance and is non-negative, gated between 0 (all channels closed) and $\bar{g}_{ion}$ (all channels open), $E_{ion}$ and $E_{leak}$ are the channel reversal potentials, $\mathcal{I}$ is the index set of intrinsic ionic currents considered in the model, and $I_{ext}$ is the current externally applied *in vitro*, or the combination of synaptic currents. Each ion channel conductance is nonlinear and dynamic, represented by $g_{ion}(V, t) = \bar{g}_{ion}m_{ion}^a(V, t)h_{ion}^b(V, t)$, where $m_{ion}$ and $h_{ion}$ are variables gated between 0 and 1, modeling the opening and closing gates of ion channels, respectively. Throughout this study, the isolated crab STG neuron model of [22] was employed.

The STG model consists of seven ion channels that operate on various time scales: fast sodium channels ($\bar{g}_{Na}$); delayed-rectifier potassium channels ($\bar{g}_{Kd}$); T-type calcium channels ($\bar{g}_{CaT}$); A-type potassium channels ($\bar{g}_A$); slow calcium channels ($\bar{g}_{CaS}$); calcium-activated potassium channels ($\bar{g}_{KCa}$); and H channels ($\bar{g}_H$).

### The calcium-homeostatic controller

The calcium-homeostatic controller adjusts all conductances using an integration rule described in [29]. Let $\bar{g}_i$ denote the conductance of ion channel $i \in [1, N]$, and $m_i$ the corresponding mRNA concentration. The dynamics of the calcium-homeostatic controller are governed by the following equations:

$$\tau_i \dot{m}_i = [Ca^{+2}]_{target} - [Ca^{+2}],$$
$$\tau_g \dot{\bar{g}}_i = m_i - \bar{g}_i,$$

where $[Ca^{+2}]$ and $[Ca^{+2}]_{target}$ represent the calcium level and its target value, respectively. In essence, the calcium level error is integrated into the mRNA levels, which subsequently modulate the corresponding channel conductances. This regulation aims to bring the calcium level to its target value, as higher channel conductances (particularly calcium-conducting ones) facilitate greater calcium influx. Starting from any initial condition, this controller drives the cell along a line in conductance space known as the homogeneous scaling line. This line passes through both the initial condition and the origin of the conductance space axes. Steady-state analysis reveals that the conductance ratios remain invariant, such that

$\frac{\bar{g}_i}{\bar{g}_j} = \frac{\tau_j}{\tau_i}$. This invariance ensures that the relative scaling of conductances is preserved during calcium-homeostatic compensation. Consequently, the cell conductance properties develop along the homogeneous scaling beam defined by these ratios.

**The neuromodulation controller**

The neuromodulation controller adjusts a subset of $n$ conductances, denoted as $\bar{g}_{mod} \in \mathbb{R}^n$ with $n < N$, using the algorithm introduced in [30]. This controller employs the concept of Dynamic Input Conductances (DICs) to determine target values for $\bar{g}_{mod}$ based on the input neuromodulator concentration, represented as $\bar{g}_0 \left([\text{nmod}]\right) \in \mathbb{R}^n$. The error between the target and the current values, $e_{mod}$, drives a Proportional-Integral (PI) controller, which updates $\bar{g}_{mod}$ to track the desired reference. The controller dynamics are described by the following equations:

$$e_{mod} = \bar{g}_0 \left([\text{nmod}]\right) - \bar{g}_{mod},$$

$$\dot{\bar{g}}_{mod} = f\left(K_p \cdot e_{mod} + K_i \cdot \int e_{mod} dt\right),$$

where $K_p$ and $K_i$ are the proportional and integral gains of the PI controller, respectively. In summary, the controller adjusts the modulated conductances $\bar{g}_{mod}$ in response to the local concentration of neuromodulator, producing new reference values for these conductances to fine-tune the cell firing pattern — effectively neuromodulating it. This neuromodulator controller produces changes in conductance ratios that are normally preserved within the calcium-homeostatic controller. The new reference values depend on the neuromodulator concentration, the type of neuron, and the current state of the cell, characterized by the full set of conductances $\bar{g}_{ion} \in \mathbb{R}^N$. The computation of these references is achieved through DICs, which encapsulate this complexity.

DICs consist of three voltage-dependent conductances that separate according to timescales: one fast, one slow, and one ultraslow, denoted as $g_f(V)$, $g_s(V)$, and $g_u(V)$, which can be computed as linear functions of the maximal conductance vector $\bar{g}_{ion} \in \mathbb{R}^N$ of an $N$-channel conductance-based model at each voltage level $V$:

$$[g_f(V); g_s(V); g_u(V)] = f_{DIC}(V) = S(V) \cdot \bar{g}_{ion},$$

where $S(V) \in \mathbb{R}^{3 \times N}$ is a sensitivity matrix that can be built by: $S_{ij}(V) = -\left(w_{ij} \cdot \frac{\partial \dot{V}}{\partial X_j} \frac{\partial X_{j,\infty}}{\partial V}\right)/g_{leak}$, where $i$ denotes the timescale, $X_j$ are gating variables of the j-th channel of the considered model and $w_{ij}$ is a timescale-dependent weight which is computed as the logarithmic distance of the time constant of $X_j$ and the timescale $i$ [40]. While the complete curve of the DICs may be of interest, only its value at the threshold voltage $V_{th}$ is used, as the values and signs of the DICs at $V_{th}$ reliably determine the firing pattern [40,68]. Thus, the following linear system $f_{DIC}(V_{th}) = S(V_{th}) \cdot \bar{g}_{ion}$ makes the link between ion channel conductances and neuronal activity.

## Supporting information

**S1 Appendix. Details and additional experiments.** We provide all the elements required to reproduce the results presented in this paper, as well as additional experiments and their results. This includes, for example, model equations and their parameter values.
(PDF)

## Acknowledgments

Use of generative AI: The ChatGPT-5 chatbot has been used to improve the syntax and grammar of several paragraphs in the manuscript. After using this tool/service, the authors reviewed and edited the content as needed and take full responsibility for the content of the published article.

## Author contributions

**Conceptualization:** Arthur Fyon, Guillaume Drion.

**Formal analysis:** Arthur Fyon, Guillaume Drion.

**Funding acquisition:** Guillaume Drion.

**Investigation:** Arthur Fyon.

**Methodology:** Arthur Fyon, Guillaume Drion.

**Project administration:** Guillaume Drion.

**Software:** Arthur Fyon.

**Supervision:** Guillaume Drion.

**Visualization:** Arthur Fyon.

**Writing – original draft:** Arthur Fyon, Guillaume Drion.

**Writing – review & editing:** Arthur Fyon, Guillaume Drion.

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
