## [Decision Letter · Decision Letter 0]

17 Feb 2026

PCOMPBIOL-D-25-02581

Controlled neuromodulation naturally interacts with homeostasis to produce robust and modulable neuronal function

PLOS Computational Biology

Dear Dr. Fyon,

Thank you for submitting your manuscript to PLOS Computational Biology. After careful consideration, we feel that it has merit but does not fully meet PLOS Computational Biology's publication criteria as it currently stands. Therefore, we invite you to submit a revised version of the manuscript that addresses the points raised during the review process.

We look forward to receiving your revised manuscript.

Kind regards,

Jian Liu

Academic Editor

PLOS Computational Biology

Daniele Marinazzo

Section Editor

PLOS Computational Biology

**Additional Editor Comments:**

The manuscript has merit, but requires significant revision to address several concerns. The biological relevance remains somewhat limited, and the authors would benefit from expanding the discussion to better connect their findings to broader biological contexts. Additionally, the analysis lacks clarity and needs more detailed explanation to guide the reader.

**Journal Requirements:**

2) Some material included in your submission may be copyrighted. According to PLOSu2019s copyright policy, authors who use figures or other material (e.g., graphics, clipart, maps) from another author or copyright holder must demonstrate or obtain permission to publish this material under the Creative Commons Attribution 4.0 International (CC BY 4.0) License used by PLOS journals. Please closely review the details of PLOSu2019s copyright requirements here: PLOS Licenses and Copyright. If you need to request permissions from a copyright holder, you may use PLOS's Copyright Content Permission form.

Potential Copyright Issues:

i) Figures 2A, and 6. Please confirm whether you drew the images / clip-art within the figure panels by hand. If you did not draw the images, please provide (a) a link to the source of the images or icons and their license / terms of use; or (b) written permission from the copyright holder to publish the images or icons under our CC BY 4.0 license. Alternatively, you may replace the images with open source alternatives. See these open source resources you may use to replace images / clip-art:

3) Please amend your detailed Financial Disclosure statement. This is published with the article. It must therefore be completed in full sentences and contain the exact wording you wish to be published.

**Reviewers' comments:**

Reviewer's Responses to Questions

**Comments to the Authors:**

Reviewer #1: The article has done intensive investigation on the neural dynamics by homeostasis and neural modulation. I have several questions and comments on this work.

1.The figures shown in Fig. 1,2,3 should have specific labels on the scales beyond the maximum values, especially we can detect how much changes are there in the x-axis and y-axis, and how much difference between different lines. Since it seems that there are so much difference between the different channel conductance, and even within the same channel (the same-colored curve), it seems in the logarithm scale there are also so much difference. So, in biological perspective, are these differences physiologically meaningful or making sense?

2.In Fig. 1 and Fig.2, it is better to show one specific example among the 200 models, how each conductance in 1 model is changing with application of neuromodulation (both sharp and controlled modulation), what’s the transient state about the spiking activity instead of the dashed dotted lines, and how the homeostasis changes each conductance, instead of showing the calcium dynamics only.

3. The difference between sharp modulation and controlled modulation is only the time period that been applied to the neuron. Then how long the controlled modulation is kind of important. What will the dynamic of the system look like if the controlled modulation withdrawn after the balance between it and homeostasis controlled have been reached? The different driving force might be mediated by the calcium homeostasis by changing other calcium dependent channel conductance, it will be necessary to clarify the tonic spiking and bursting behavior by detailed dynamic analysis with the different combinations of stimuli. Minor: the burstiness in Hz shown in the Fig.c is also logarithm? The detailed scale should be marked on the y-axis; how do you define the frequency? The inter-burst-interval? Or the spike frequency within one burst?

4. in Fig. 4, how the black curve happened? The title in A: Homeostasis controller “alone” with sharp neuromodulation, might be “along”?

5. In Fig. 5 the y-axis should be labeled with different scale such that readers will know how much frequency they are distributed in. Can the author give specific explanations on why the different channel type blockage leads to different physiological and compensational results?

6. In Fig. 6, the Fig.A x-axis with multiplication of different numbers, what does that mean?

Also, the neuromodulation seems only change the slow rhythm, since there is a red neuron connecting between fast rhythm and slow rhythm, how strong the neuromodulation will change the purple and blue neuron firing activates and which factors will affect the patterns that system could generate?

Reviewer #2: This paper introduces the concept of “controlled neuromodulation,” contrasted with “sharp neuromodulation,” and demonstrates its better compatibility with neuronal homeostats. I found the manuscript well written and the scientific ideas interesting and original. However, I have several major concerns that, if addressed, would substantially improve the quality of the manuscript making it suitable for publication.

Major concerns

1: In the current form, the manuscript describes the behavior of models under controlled neuromodulation more than delineating "inner workings" by which controlled neuromodulation is implemented. I understand that the method has been published previously (ref. 30) and that additional details are provided in the Methods section. Nevertheless, I recommend adding more explanation in the main text of how controlled neuromodulation operates, ideally through concrete examples and intuitive descriptions that would help a broad readership.

For example, though I do find Figure 4 helpful, it also primarily focuses on model behavior. It may be more informative to show how distinct components of the model (e.g., feedforward vs. feedback) contribute to achieving the model behavior.

2: I am not yet fully convinced that controlled neuromodulation, as defined here, can be realized by known biological mechanisms. The manuscript would benefit from concrete proposals or examples of signaling pathways that could plausibly implement controlled neuromodulation.

3: I find Figure 5 and the related discussion quite amazing, but they raise questions, especially in relation to Concern 2, about what limits such compensation in real neurons. If compensation can be pushed arbitrarily far as in the model, there should be no neurological and/or neurodegenerative conditions caused by ion-channel deficiencies or malfunctions as they can be compensated. In that regard, I am not entirely sure what the authors intend to deliver with Fig. 5.

Minor comments

1. Page 5, lines 90–92 (“to facilitate … original implementation”) The meaning of this phrase is unclear to me.

2. Page 6, line 124 (“cyclic amp”) Please spell out “cyclic AMP (adenosine monophosphate)” at first use.

3. Could the authors comment on how controlled neuromodulation can be extended to work in the context of dendritic conductances and compartmentalization?

Reviewer #3: Summary:

The manuscript ‘Controlled neuromodulation naturally interacts with homeostasis to produce robust and modulable neuronal function’, by Fyon and Drion, studies the interaction of a calcium homeostatic controller with a neuromodulatory controller of ion channels in the classical STG neuron model by Liu et al., 1998. The calcium homeostatic controller is based on O’Leary et al., 2014 that models changes to the ion channel conductance to match an intrinsic calcium ion concentration set-point. To this intrinsically controlled ion channel model of a neuron, the authors apply a neuromodulatory perturbation, theorized to instantaneously change the conductances of two ion channels (A-type K and slow-Ca). This perturbation is applied as i) a sudden step change or as ii) an orchestrated neuromodulatory controller (based on Fyon, Franci and Drion, 2023) interacting with the intrinsic calcium homeostatic controller.

In a computational model of a population of STG neurons, the authors study the progression of the burstiness of the neuron in response to the applied neuromodulator, and the evolution of the ion channel conductances over time (Fig 1, 2 and 3). Some insight on how the two controllers interact is also presented (Fig 4). The authors also employ virtual knock-outs to identify channels critical for restoring firing properties (Fig 5) and test the model within a central pattern generator (CPG) network (Fig 6). They demonstrate that neuromodulatory sensitivity is vital for establishing slow rhythms and that interacting controllers outperform sudden perturbations.

The study presents a framework to study neuromodulation in the broader context of homeostatic controllers, and addresses how a modelled neuron might deal with degeneracy. As such, it offers a valid approach and offers new insights into the challenges a biological neuron must endure while retaining its function.

The manuscript is well-written and the methodology is sound; the availability of code to run the simulations presented is appreciated. However, the presentation of the data in the figures requires significant aesthetic and technical refinement to clearly convey the message (see detailed comments below).

I have two major comments and several minor comments to improve this MS.

Major comments:

1 Ih Currents:

It is unclear as to why the neuromodulatory response, as the authors have modelled, instantly changes two ion channel types (A-type K and Slow Ca type) and not the Ih-channels.

Ih-current carrying ion channels are well known to be sensitive to cAMP levels: When cAMP binds to the channel, it shifts the activation curve of the channel to more positive voltages. As a result, the channel opens faster and at less negative voltages.

Given that the Ih currents set the pacemaker currents in the rhythmic nature of the spiking, it seems natural to include the instantaneous changes to these channels in this context.

Please justify this omission or include this in your study.

2. Temporal dynamics:

Why is the neuromodulatory perturbation not relaxing after a while? [nmod] is presented, and a change of a dynamical interaction is seen, but this assumes that the [nmod] persists for long - necessarily longer than the two homeostatic mechanisms to kick in.

If the neuromodulator doesn't wash out, the homeostasis isn't responding to a perturbation; it's responding to a permanent change in the environment. Most neuromodulators would have some adaptation even. One would anticipate that the A-type conductivity and the CaS would return naturally to their previous values once the neuromodulation is washed away.

Please justify why this is modelled this way, or include this in your study.

Minor comments:

1. Overall, all the figures could be much better.

Figure 1.

An equivalent to Fig2A could be placed here.

The top row (unlabeled, uncaptioned) represents three exemplar traces from the population at three different time points (unlabeled, not mentioned, correspond to dashed vertical lines?).

Middle row, consider showing the trace of one neuron and showing the rest as histograms of population at the vertical dashed lines.

[nmod] is a notation that is only accessible after reading the methods. Please consider a better label for the perturbation.

Include the vertical dashed lines in the bottom row for the ca-conc.

Figure 2.

Same as for Figure 1.

The schematic could be improved.

Figure 3.

Perhaps indicate the time points from Figure 1,2 here for the A, B.

Consider showing the trajectories for fewer neurons, and show the conductivity of the population as a histogram of before and after on the x and y axis.

Subplot C, has the inset plot Burstiness (Hz) [y-axis] versus X-axis (is it time?): this is totally unclear what you wish to show here. I realize that 65% implies that 65% of the neuron models are bursting (what happens to the rest) at the end. But what is the raster (raster-like) plot supposed to show?

The labels ‘Initial condition from previous step’ and ‘final condition from current step’ could be improved.

Figure 4.

While I recognize the value of the insight from this schematic. I fear it is misrepresenting the sharp neuromodulation (subplot A). Based on Figure 1 bottom row, the Ca-levels are restored to the target Ca level. Please clarify this.

In A. what is the dashed arrow representing?

The labels could be better. There is too much text.

Could this plot be generated from the actual simulations instead?

Figure 5.

The raster (raster-like) plots are unclear. What is the X-axis? What is the Yaxis? Frequency of burstiness?

There is redundant labeling ‘Control condition’ ‘After blockage’, ‘With compensation’

Figure 6.

Subplot A, right side: what do the colors here mean?

There are some interesting results here, but they are rushed. Please consider elaborating on these.

2. Please consider taking the section on DA neuron out of the main MS entirely. It feels out of place. Perhaps you can consider writing it as another case study of this mechanism and mention it only in the discussion.

3. Throughout the work, ‘homeostatic controller’ is used. I urge you to be specific about it and call it a calcium-homeostatic controller all through. There can be many other homeostatic controllers, in your work you re-implemented O’Leary’s ca set-point homeostatic controller. In one way, your work is also a homeostatic controller that combines ca conc and nm response.

4. Please consider changing the names of what you call [nmod] followed by [sharp neuromodulatory controller] versus an [interacting neuromodulatory controller]. These notions are difficult to follow. You could revise the naming conventions to better reflect on what your modelling effort is trying to achieve. Perhaps considering your work as a nested feedback loop, that is over the ca-homeostasis would be easier to interpret.

5. Please change the title to better reflect what the paper is showcasing. Something in the lines of ‘Interplay between calcium-mediated homeostasis and neuromodulatory control in STG neurons guarantees robustness’.

**Have the authors made all data and (if applicable) computational code underlying the findings in their manuscript fully available?**

The PLOS Data policy requires authors to make all data and code underlying the findings described in their manuscript fully available without restriction, with rare exception (please refer to the Data Availability Statement in the manuscript PDF file). The data and code should be provided as part of the manuscript or its supporting information, or deposited to a public repository. For example, in addition to summary statistics, the data points behind means, medians and variance measures should be available. If there are restrictions on publicly sharing data or code —e.g. participant privacy or use of data from a third party—those must be specified.requires authors to make all data and code underlying the findings described in their manuscript fully available without restriction, with rare exception (please refer to the Data Availability Statement in the manuscript PDF file). The data and code should be provided as part of the manuscript or its supporting information, or deposited to a public repository. For example, in addition to summary statistics, the data points behind means, medians and variance measures should be available. If there are restrictions on publicly sharing data or code —e.g. participant privacy or use of data from a third party—those must be specified.requires authors to make all data and code underlying the findings described in their manuscript fully available without restriction, with rare exception (please refer to the Data Availability Statement in the manuscript PDF file). The data and code should be provided as part of the manuscript or its supporting information, or deposited to a public repository. For example, in addition to summary statistics, the data points behind means, medians and variance measures should be available. If there are restrictions on publicly sharing data or code —e.g. participant privacy or use of data from a third party—those must be specified.requires authors to make all data and code underlying the findings described in their manuscript fully available without restriction, with rare exception (please refer to the Data Availability Statement in the manuscript PDF file). The data and code should be provided as part of the manuscript or its supporting information, or deposited to a public repository. For example, in addition to summary statistics, the data points behind means, medians and variance measures should be available. If there are restrictions on publicly sharing data or code —e.g. participant privacy or use of data from a third party—those must be specified.

Reviewer #1: Yes

Reviewer #2: Yes

Reviewer #3: Yes

PLOS authors have the option to publish the peer review history of their article (what does this mean?). If published, this will include your full peer review and any attached files.). If published, this will include your full peer review and any attached files.). If published, this will include your full peer review and any attached files.). If published, this will include your full peer review and any attached files.

...

Reviewer #1: No

Reviewer #2: No

Reviewer #3: No

**Figure resubmission:**
---

## [Decision Letter · Decision Letter 1]

28 Mar 2026

Dear Dr Fyon,

We are pleased to inform you that your manuscript 'Activity-dependent neuromodulation and calcium homeostasis cooperate to produce robust and modulable neuronal function' has been provisionally accepted for publication in PLOS Computational Biology.

Best regards,

Jian Liu

Academic Editor

PLOS Computational Biology

Daniele Marinazzo

Section Editor

PLOS Computational Biology

Please consider the minor comment regarding Figure 3 for the final version.

Reviewer's Responses to Questions

**Comments to the Authors:**

Reviewer #1: The authors have made great effort to improve the manuscript and all questions are clarified.

Reviewer #2: In this revision, my previous major and minor comments are adequately addressed. I am also impressed by new material that provides stronger support to the study. I am happy to recommend publishing this article.

Reviewer #3: Thank you for addressing my comments and for incorporating my suggestions.

I appreciate the revision. My only remaining (and minor) point is that Figure 3 feels aesthetically dated. I’ll leave it to the authors and editors to decide if a visual refresh is warranted before publication.

**Have the authors made all data and (if applicable) computational code underlying the findings in their manuscript fully available?**

The PLOS Data policy requires authors to make all data and code underlying the findings described in their manuscript fully available without restriction, with rare exception (please refer to the Data Availability Statement in the manuscript PDF file). The data and code should be provided as part of the manuscript or its supporting information, or deposited to a public repository. For example, in addition to summary statistics, the data points behind means, medians and variance measures should be available. If there are restrictions on publicly sharing data or code —e.g. participant privacy or use of data from a third party—those must be specified.requires authors to make all data and code underlying the findings described in their manuscript fully available without restriction, with rare exception (please refer to the Data Availability Statement in the manuscript PDF file). The data and code should be provided as part of the manuscript or its supporting information, or deposited to a public repository. For example, in addition to summary statistics, the data points behind means, medians and variance measures should be available. If there are restrictions on publicly sharing data or code —e.g. participant privacy or use of data from a third party—those must be specified.requires authors to make all data and code underlying the findings described in their manuscript fully available without restriction, with rare exception (please refer to the Data Availability Statement in the manuscript PDF file). The data and code should be provided as part of the manuscript or its supporting information, or deposited to a public repository. For example, in addition to summary statistics, the data points behind means, medians and variance measures should be available. If there are restrictions on publicly sharing data or code —e.g. participant privacy or use of data from a third party—those must be specified.requires authors to make all data and code underlying the findings described in their manuscript fully available without restriction, with rare exception (please refer to the Data Availability Statement in the manuscript PDF file). The data and code should be provided as part of the manuscript or its supporting information, or deposited to a public repository. For example, in addition to summary statistics, the data points behind means, medians and variance measures should be available. If there are restrictions on publicly sharing data or code —e.g. participant privacy or use of data from a third party—those must be specified.

Reviewer #1: Yes

Reviewer #2: Yes

Reviewer #3: Yes

PLOS authors have the option to publish the peer review history of their article (what does this mean?). If published, this will include your full peer review and any attached files.). If published, this will include your full peer review and any attached files.). If published, this will include your full peer review and any attached files.). If published, this will include your full peer review and any attached files.

...

Reviewer #1: No

Reviewer #2: No

Reviewer #3: No

---

## [Editor Report · Acceptance letter]

PCOMPBIOL-D-25-02581R1

Activity-dependent neuromodulation and calcium homeostasis cooperate to produce robust and modulable neuronal function

Dear Dr Fyon,

I am pleased to inform you that your manuscript has been formally accepted for publication in PLOS Computational Biology. Your manuscript is now with our production department and you will be notified of the publication date in due course.

With kind regards,

Anita Estes
